# Co-Evolution as more than a scalable alternative for Multi-Agent Reinforcement Learning

## Abstract

In recent years, gradient based multi-agent reinforcement learning is growing in success. One contributing factor is the use of shared parameters for learning policy networks. While this approach scales well with the number of agents during execution it lacks this ambiguity for training as the number of produced samples grows linearly with the number of agents. For a very large number of agents, this could lead to an inefficient use of the circumstantial amount of produced samples. Moreover in single-agent reinforcement learning policy search with evolutionary algorithms showed viable success when sampling can be parallelized on a larger scale. The here proposed method does not only consider sampling in concurrent environments but further investigates sampling diverse parameters from the population in co-evolution in joint environments during training. This co-evolutionary policy search has shown to be capable of training a large number of agents. Beyond that, it has been shown to produce competitive results in smaller environments in comparison to gradient descent based methods. This surprising result make evolutionary algorithms a promising candidate for further research in the context of multi-agent reinforcement learning.

## 1 Introduction

The core idea of this work is based on a union of the concepts of parameter sharing for multi-agent policies and policy search with evolutionary algorithms (EA). In general stochastic gradient descent (SGD) together with back-propagation is a powerful approach for optimizing neural network parameters. For on-policy reinforcement learning with SGD the generated samples are dependent on the current policy which is again subject to the gradient update. This circumstance makes the vanilla policy gradient high in variance and slow in learning Sutton et al. (1999). Hence contemporary policy gradient methods use baseline terms and other remedies to reduce the variance and further increase sample efficiency. One exemplary algorithm of this class is PPO (Schulman et al., 2017b) which further includes a clipping based on the probability ratio of the old and updated policy.

Policy search with evolutionary algorithms is gradient-free and its use in reinforcement learning has already a long history (Heidrich-Meisner & Igel, 2008; Moriarty et al., 1999). One advantage of this method is the lack of backpropagation and its computational cost that scales linearly with the amount of samples per iteration. One disadvantage of evolutionary algorithms, as black-box optimization methods, is the absence of step-wise information about state, action, and reward transitions. Furthermore, a key distinction to on-policy policy gradient methods is that a ensemble of policies, the population, is evaluated in each iteration. This increases the demand for necessary samples for each iteration compared to policy gradients. However, due to the advent of cloud computing and multi-threading CPU architectures parallel sampling became more accessible to a broader audience.

Another difficulty of evolutionary algorithms was that they did not scale well with the number of parameters needed for deep learning. One efficient and almost hyperparameter free algorithm CMA-ES Hansen et al. (2003) scales unfortunately quadratically with the amount of parameters. Nevertheless small networks with just $10^4$ parameters have shown to be capable of learning vision-based tasks as shown by Tang et al. (2020). However, that deep reinforcement learning is possible with evolutionary algorithms is shown by multiple works that use some variants of evolutionary strategies

(ES) or genetic algorithms (GA). In particular, the work of Salimans et al. (2017) has validated the capabilities of ES in deep reinforcement learning. Some of their core outcomes were that policy search with ES is relatively easy to scale with almost linear training time reduction. Since the policy update time cost is relatively cheap in contrast to the sampling time. Further, in their experiments the data efficiency was not substantially worse, with a range of 3-10 times of compared policy gradient methods. Another noteworthy implementation of ES in deep reinforcement learning is from Conti et al. (2017) that extends the fitness objective by a novelty term (Lehman & Stanley, 2011) for improving exploration. Besides the work of Such et al. (2017) has shown that also GA are capable of training large networks for more complex tasks like Atari games and continuous control problems. Additionally, they showed a method describing network parameters in a compressed form through a chain of mutation seeds. Moreover, in the contribution of Hausknecht et al. (2014) one subclass of GA the generative encoding algorithm HyperNEAT (Stanley et al., 2009) was used. It was besides DQN (Mnih et al., 2013) one of the first solutions for Atari (Brockman et al., 2016) games. While not too popular in single-agent reinforcement learning, policy search with EA still showed some competitive results over recent years.

Multi-agent reinforcement learning (MARL) is an interesting extension of reinforcement learning that involves game theoretical issues. The advantages of MARL compared to RL are obvious for problems that can be separated into multiple agents. The separation of state and action is limiting the curse of dimensionality and it enables further decentralized learning and execution schemes compared to the single agent case. Nevertheless, MARL introduces additional problems such as the non-stationarity caused by interdependent non-converged policies. Some other important problems dependent on the specific task are the credit attribution, heterogeneity of agents, partial-observable states and communication between agents. The issue of scaling the number of agents is the primary motivation for this work. The study of (Gupta et al., 2017) investigated the concept of parameter sharing (PS) for policies. This method can scale in execution to an arbitrary amount of agents if the problem itself does not change due to the number of agents. Else curriculum learning Bengio et al. (2009) was found to enable adjustment to tasks that change with size. Yet not every environment allows scaling in size and considering very large environments an overabundance of samples generated per timestep and agent could further slow down SGD. Still for the agent-wise smaller 5 vs. 5 player game of DOTA 2 the work of (Berner et al., 2019) showed that also learning on a larger scale with batches of about $2 \times 10^6$ timesteps every 2 seconds is feasible for parameter sharing PPO in self-play.

## 2 METHODS

First, let us revisit evolutionary strategies (ES) as a particularly interesting evolutionary algorithm for this work. For ES the population is sampled from a normal distribution with its mean and variances as only describing parameters. For parameter distribution based RL the authors of REINFORCE (Williams, 1992) already described a gradient ascent term. For this gradient ascent term similar to TRPO (Schulman et al., 2017a) a natural gradient can be defined that accounts for the KL-divergences (Wierstra et al., 2014). A simple but empirical still effective variant of ES with a constant variance is that of Salimans et al. (2017). This makes that variant a good candidate for large-scale MARL experiments. The work of Lehman et al. (2018) gives an insightful description of the differences in learning between ES and finite difference based methods. Further, they conclude to the understanding that because ES optimizes the average reward of the population they seek especially robust parameters for perturbation. Further, it is assumed by the authors that this parameter robustness could be a useful feature for co-evolution and self-play. The pressure to find a distribution that finds samples compatible for co-operation could lead to more stable solutions. Some successful demonstrations of early self-play with EA is found in Chellapilla & Fogel (2001).

All in all, investigating co-evolution on MARL problems seems interesting. But how can co-evolution be realized in the context of collaborative MARL? Parameter sharing is a valid option for policy gradient methods and would be also a seemingly viable option for policy search within EA. Yet EA need to sample multiple policies for their expected fitness values. This increases sampling demand linearly by population size. By co-evolutionary sampling, one could reduce this growing complexity, especially for environments with many agents. However, if no local rewards are available credit assignment could be a problem and additional runs in different pairings are necessary for a good estimation of the fitness.

How to pair solutions for co-evolutionary fitness evaluation in environments is an open question. The here proposed method should be simple. It should work disregarding the chosen EA algorithm, the number of the population, evaluation workers and agents in the environment. The Algorithm 1 depicts such a simple co-evolutionary method (COMARL). Here for each episode that will be run by an evaluation worker a random set of the population is drawn. The only requirement is that the number of total runs per iteration should be minimal and that for each population member at least $\eta$ episodic return samples were generated.

So with COMARL, there is a simplistic co-evolutionary pairing scheduler that should work for every EA in training. But how should be a solution selected and executed from co-evolutionary training? A method that would work for every EA would be to use the solution with the best sampled fitness during training and execute it as a parameter sharing policy. However, this does not consider the performance of other solutions. Sometimes rewards in MARL can be negatively correlated and the sampled fitness is only good because of the failure of other agents. Moreover, other solution selection methods could be specific for some EA. In the case of ES, the population with maximum average fitness could be considered. Here the population defining mean in parameter sharing or samples from the parameter distribution could be the selected solutions for the policies in execution. This could limit the variance in returns of agents with respect to the first proposed method.

---

**Algorithm 1:** Simple Co-Evolutionary MARL (COMARL)

---
**Input:**
EA solver $\mathcal{E} : \{F(\theta_0), \ldots, F(\theta_j)\} \to \{\theta_0, \ldots, \theta_j\} = population \ \mathbf{p}$,

episodic return evaluation $\mathcal{F}^{(i)} : \{\theta_m, \ldots, \theta_n\} = \Theta^{(i)} \to \{r_m^{(i)}, \ldots, r_n^{(i)}\} = \mathbf{r}^{(i)}$,

minimum samples $\eta$

initialize all $\theta_i \in population \ \mathbf{p}$ according to EA method;

**while** *genration is not max_generations* **do**

> Create the minimum amount of random shuffled multiset subsets $\Theta^{(i)}$ of set $\mathbf{p}$ such that $\dim(\Theta^{(i)}) = \dim \mathbf{r}^{(i)}$ and such that each element of the population $\mathbf{p}$ occurs at least $\eta$ times ;
>
> Evaluate all $\mathcal{F}^{(i)}$ ;
>
> Calculate fitness from average episodic returns $F(\theta_l) = average\left(r_l^{(i)}\right)$ ;
>
> Update population $population \leftarrow \mathcal{E}(F(\theta_0), \ldots, F(\theta_j))$ ;

**end**

---

## 3 EXPERIMENTS

The issue of scaling the number of agents is the initial incentive for this paper. This part will be discussed in the experiments of 3.2. How the co-evolutionary algorithm compares to SOTA algorithms in smaller benchmark environments is covered in 3.3.

### 3.1 SETUP

For the experiments about scalability in 3.2 the Longroad environment has been used which is loosely based on traffic control with circular interweaving crossroads. It has discrete action and partial observable discrete state space and can simulate thousands of agents with multiple frames per second on a single machine. The agents control the traffic either by blocking one road or un-blocking the opposite direction. For each waiting vehicle a negative local reward is given. Otherwise to promote cooperative behavior a positive global reward is given for each continuously moving vehicle. The goal of this environment is not showcase any sophisticated learning behavior but rather to be a primitive test bed for testing out large scale MARL concepts.

Since the environment used for the scaling experiments was not used in any other publications it would be interesting to investigate the co-evolutionary method in more established MARL environments. Here the SISL environments first created for Gupta et al. (2017) and now maintained as part of PettingZoo (Terry et al., 2020) could give more comparability with the experiments conducted in 3.3. Unfortunately, it seems that the Waterworld environment is not supported in newer releases which is why it is omitted in the following experiments. However, with Pursuit as a discrete environ-

ment and Multiwalker as a continuous control task, some diversity is already considered. Further, a slightly self-patched version of the newest release of PettingZoo was used to enable compatibility with newer NumPy versions.

The earlier co-evolutionary experiments with COMARL of 3.2 are based on the EA implementations of Ha (2017). This includes a CMA-ES implementation based on the pycma (Hansen et al., 2019) library, a ES implementation of Salimans et al. (2017), a implementation of the natural evolutionary strategy variant of PEPG (Sehnke et al., 2010) and also a simple genetic algorithm. The co-evolutionary experiments with COMARL conducted in 3.3 are based on a simpler re-implementation with JAX (Bradbury et al., 2018) and FLAX (Heek et al., 2020) of the fixed variance ES algorithm of Salimans et al. (2017) in the toolbox of Ha (2017). Where applicable for stochastic gradient ascent of the parameter distribution the Adam (Kingma & Ba, 2017) optimizer was used. The reference trials were conducted in both with RLlib (Liang et al., 2018) and their implementations of the on-policy PPO (Schulman et al., 2017b), off-policy Ape-X DQN/DDPG (Horgan et al., 2018) and parameter sharing Gupta et al. (2017). For the experiments in the Longroad environment, a MLP with hidden layer shape $(16, 16)$ was used for the EA and $(256, 256)$ for PPO. All methods in 3.3 use a MLP with hidden layer of shape $(64, 64)$. The parallelization of sampling was realized with Ray (Moritz et al., 2018). For details of the chosen hyperparameters consult Appendix A.1.

For the co-evolutionary computation in 3.2 an AMD Ryzen 5900X CPU was used with 4-10 workers for parallel sampling and for the reference algorithms a NVidia Quadro K620 GPU was used. The devices for 3.3 were an AMD Ryzen Threadripper PRO 3955WX CPU for the co-evolutionary method with 30 concurrent sampling workers and for the reference algorithms an Intel XEON E-2278G CPU with 15 sampling workers and a Nvidia RTX 3080 GPU for inference and training.

## 3.2 SCALABILITY

Rather than examining the performance of co-evolutionary learning the experiments in this category should show its feasibility for a growing number of agents in contrast to single-agent and parameter sharing policy gradient methods. Beginning with 10 agents, the three evolutionary algorithms Genetic Algorithm (GA), CMA-ES and OpenAI-ES (Salimans et al., 2017) and the centralized single agent PPO and parameter sharing PPO (PPO-PS) are evaluated. The length of an episode is 50 timesteps. For COMARL each policy solution is sampled 16 times with a total of 400 episodes per fitness evaluation. The policy gradient methods have a batch size of $80 \times 50$ timesteps. All algorithm types are run for 500 iterations. The Figure 1a shows the learning curve for the averaged total mean reward over three trials. The PPO-PS performs fast and sample-efficient on the first iterations but the slower centralized single agent PPO ultimately exceeds the results of the PPO-PS runs. All three EA in COMARL perform worse than the policy gradients. Still, for this number of agents all methods are feasible.

The second experiment shown in Figure 1b scales the number of agents to 100. Here the number of samples per episode for parameter sharing PPO-PS grows ten-fold. So does its learning time consumption when still all produced samples should be considered. The curve for PPO-PS thus only depicts a single incomplete trial which seems to perform still better than the other methods. Seemingly the curse-of dimensional affects the single-agent PPO as no increase in return can be observed. Considering learning regarding simulated timesteps the co-evolutionary methods seem to profit from the increased number of samples per timestep.

Ultimately the trials for 1000 agents are depicted in Figure 1c. Because of the expected learning time consumption of PPO-PS for this setting it was omitted for this number of agents. Unsurprisingly the single-agent PPO did not overcome the curse of dimensionality with growing state and action space. Again the co-evolutionary methods seem to benefit from the increased number of samples per simulation timestep.

To stress test the capabilities of COMARL a single trial with $10^6$ agents was conducted using OpenAI-ES as underlying EA as shown in Figure 2. The iterations for this example consist of only one episode per worker. Still for such a high number of agents, good progress in learning could be observed. Since the parameter update time of the evolutionary algorithms just depends on the size of the population, the parameter space and their hyperparameter it is indifferent to the number of samples needed for fitness evaluation. The fitness evaluation becomes the bottle-neck in that case as the Table 3 and Table 4 further illustrate.

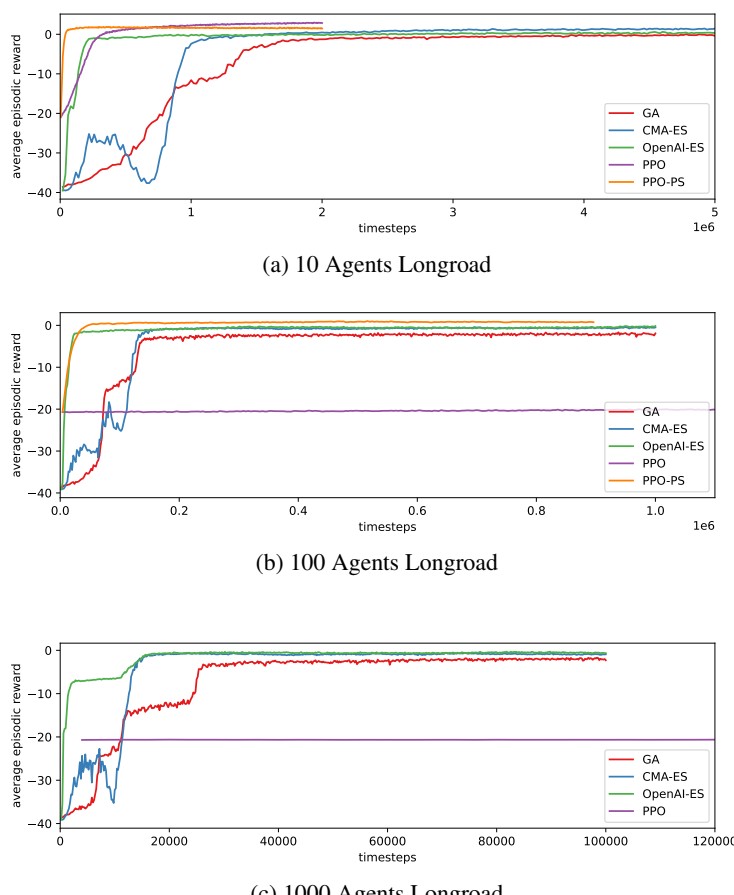

(a) 10 Agents Longroad

(b) 100 Agents Longroad

(c) 1000 Agents Longroad

Figure 1: Averaged learning curves for different scaled Longroad environments. The curves are the averaged episodic mean over all trials. The single-agents PPO reward is normalized by the number of agents. For Figure 1b and Figure 1c the PPO and PPO-PS curves depict only a single trial

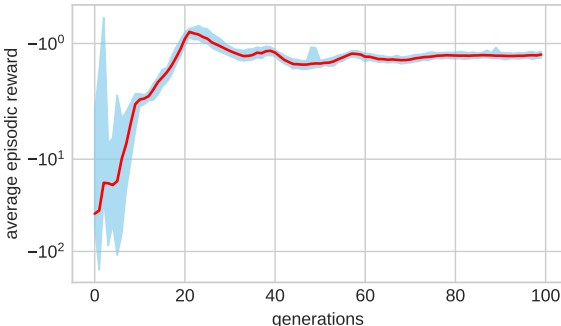

Figure 2: Learning curve in logarithmic scale for $10^6$ agents in Longroad with co-evolutionary OpenAI-ES. The red line gives the average episode reward over the population and the blue area the range of episodic rewards for that generation.

## 3.3 COMPARABILITY

The preceding experiments of 3.2 have shown that the co-evolutionary method can scale to numerous agents. However, the Longroad environment is not a good benchmark in a general setting. Hence the following experiments examine the co-evolutionary method in the for MARL research more prevalent SISL environments, Pursuit and Multiwalker, to give an idea of the performance in comparison. For each environment and method 3 trials on the same set of seeds was run.

For the continuous control problem of Multiwalker as Figure 3 depicts the co-evolutionary variant of OpenAI-ES shows a stable performance over all trials. Considering the wall clock-time it reaches very fast a region of positive rewards and is seemingly also sample-efficient in contrast to parameter sharing Ape-X DPPG which shows also more variance in its outcome on different trials. Under this setting parameter sharing PPO fails to learn the task at hand. With the related method of parameter sharing TRPO better results were accomplished in Gupta et al. (2017). But the question arises of how comparable the new versions of Multiwalker are to that examined in older work.

In the discrete environment of Pursuit, the results of the investigated methods are more aligned as Figure 3 shows. Parameter sharing Ape-X DQN is the only algorithm to reach positive rewards on a trial but is again variant in its outcome for different runs similar to Ape-X DPPG in Multiwalker. Else parameter sharing PPO seems to be more sample efficient while co-evolutionary OpenAI-ES is slightly faster on wall-clock time. However, the difference is not too significant.

Concerning the original work of OpenAI-ES (Salimans et al., 2017) in the single-agent domain that achieved sample-efficiencies of 3-10 times to reference non-EA algorithms, the results for the SISL environments are surprisingly far from that range. Still, the scale of parallelization differs significantly from this work which could be one factor.

The solution selection method of choosing the parameters with the best sampled fitness over all generations was evaluated in parameter sharing execution for 32 episodes. The detailed results are shown in Table 6. This method was chosen because it works with all EA. However the results are not overwhelming. Only for the Multiwalker 2 of 3 trails were close to the result of the best sampled fitness. Else the evaluation of the solution was worse than the mean return of its population during training.

Moreover, for both environments the throughput for sampling time steps is greater by one order of magnitude for the co-evolutionary method compared to PPO. This could be because the inference is run on multiple CPU threads instead of a single GPU. Further for updating the parameter distribution with OpenAI-ES less than 2% of total time was utilized while for parameter sharing PPO the policy update consumed about 80% of the total time. See Table 5 for actual numbers.

Since sampling fitness for a population or parameter distribution with parameter sharing was deemed ineffective it was not further investigated. Nevertheless, the next experiment compares it to co-evolutionary sampling for a single trial in the Multiwalker environment since it has only 3 agents

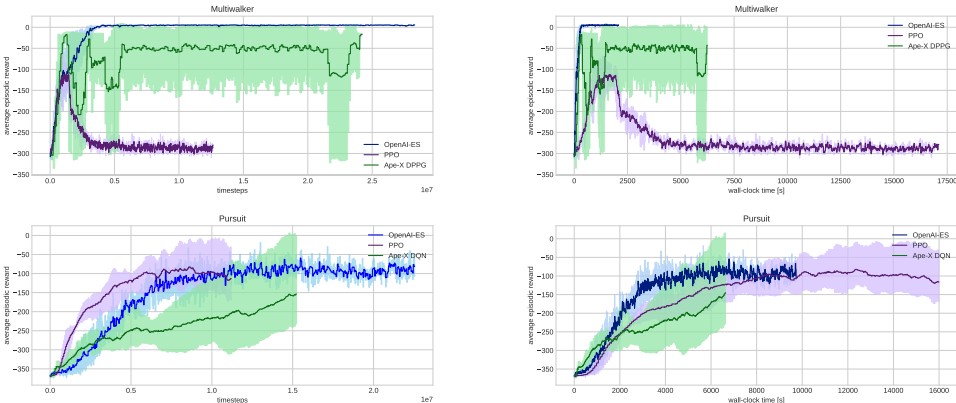

Figure 3: Averaged learning curves of the examined SISL environments regarding time and sample efficiency. The transparent area shows the range of the mean episodic return for all trials.

for its regular configuration. Figure 4 depicts the fitness ascend for the whole population and its statistics. For parameter sharing 10 complete episodes were sampled while for COMARL it was just 10 agent-wise experiences among all evaluated episodes each iteration. This already leads to a thrice as high sampling time for learning with parameter sharing for a parameter distribution (Table 7. In this example, the deviation between fitness values is higher for COMARL than PS and it reaches further a higher fitness plateau. Regardless, for just a single trial nothing conclusive about the performance in comparison to parameter shared training can be shown except its timings. Anyway, after the experiments in section 3.3 and 3.2 co-evolution seems to be also a viable option for a smaller number of agents and practically the only one for a very high number of agents.

## 4    CONCLUSION AND FUTURE WORK

The motivation behind this work was to find a method that scales to a high number of agents. The here proposed co-evolutionary method achieved this for large scale environments in the range of up to $10^6$ agents. Since the investigated large scale environment is very simplistic more meaningful and realistic environments in that range of agents could be examined next. Moreover the compromise of information lost in this black-box optimization scheme for policy search improves high scaling and decentralization of sampling. Here also EA with asynchronous or decentralized updates could be of particular interest for MARL.

Unexpected was the competitive result in a small comparative study to SOTA on and off-policy parameter sharing methods. Here the fixed variance natural evolutionary strategy (Salimans et al., 2017) in co-evolution showed still good data-efficiency. Further an exhaustive comparative study in more MARL environments could be of interest. A lack of well established benchmark environments in contrast to single agent reinforcement learning is an ongoing problem. Since it is assumed that co-evolution could help with robustness research in task that involve for instance adversarial behavior of agents could be of interest.

Moreover, this work focused on finding a co-evolutionary training method regardless of the underlying EA. For evaluation a solution for execution has to be defined. Here further research with analytical focus on specific EA and a solution selection methods for them could be of interest. Further the co-evolutionary pairing itself could exhibit improvement in alignment with specific EA.

In conclusion it is hoped that this work can spark interest in policy search with evolutionary algorithm in the context of multi-agent reinforcement learning.

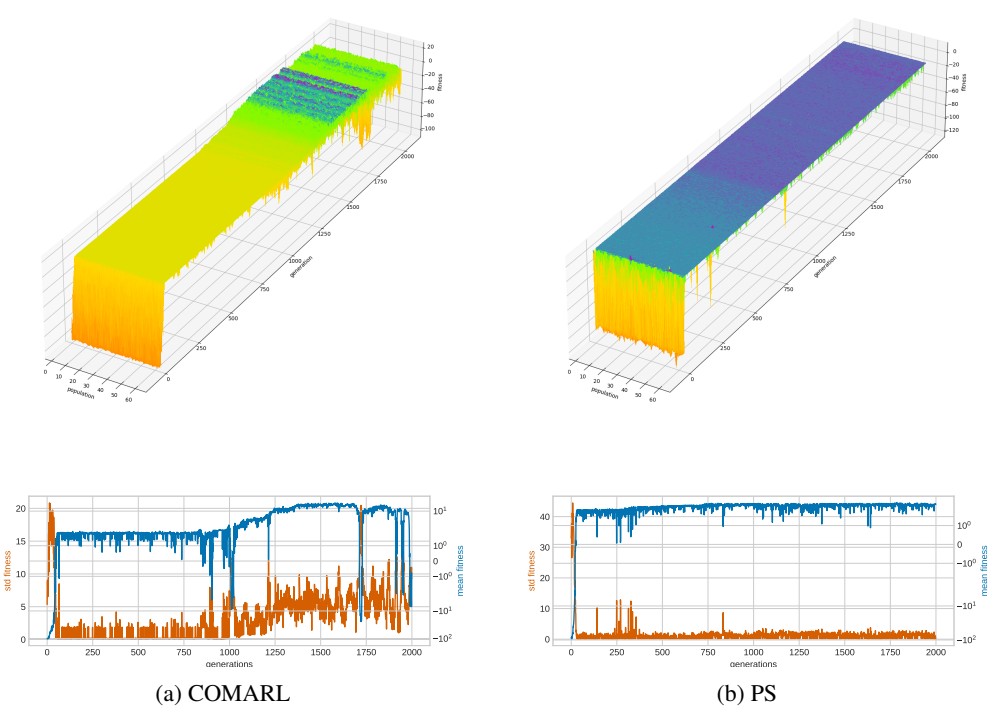

(a) COMARL
(b) PS

Figure 4: Fitness plot and statistics for Multiwalker with OpenAI-ES. The 3D plot above shows the fitness for each member of the population respectively for COMARL in Figure 4a and parameter sharing in Figure 4b. Correspondingly below the statistics to the mean and standard deviation of the fitness are given. The axis of the mean is logarithmic and its zero level shifted.

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

# A APPENDIX

## A.1 HYPERPARAMETER

### A.1.1 SCALABILITY

Table 1: Hyperparameter of the EA and policy gradient algorithms in 3.2

(a) OpenAI-ES

| Parameter | Value |
|---|---|
| Feedforward shape | $(16, 16)$ |
| initial variance | 0.1 |
| variance decay | 0.9999 |
| variance limit | 0.02 |
| learning rate | 0.1 |
| weight decay | 0.005 |

(b) CMA-ES

| Parameter | Value |
|---|---|
| Feedforward shape | $(16, 16)$ |
| initial variance | 0.1 |

(c) Genetic Algorithm

| Parameter | Value |
|---|---|
| Feedforward shape | $(16, 16)$ |
| elite ratio | 0.1 |
| variance | 0.5 |
| weight decay | 0.005 |

(d) PPO

| Parameter | Value |
|---|---|
| Feedforward shape | $(256, 256)$ |
| activation function | tanh |
| learning rate | 0.00005 |
| discount factor | 0.99 |
| clipping factor | 0.3 |
| entropy coefficient | 0.0 |
| batch size | 80 x 50 |

A.1.2   COMPERABILITY

Table 2: Hyperparameter of the EA and reference algorithms in 3.3

(a) OpenAI-ES

| Parameter | Value |
|---|---|
| Feedforward shape | $(64, 64)$ |
| initial variance | 0.1 |
| variance decay | 0.9999 |
| variance limit | 0.02 |
| learning rate | 0.1 |
| weight decay | - |

(b) PPO

| Parameter | Value |
|---|---|
| Feedforward shape | $(64, 64)$ |
| activation function | tanh |
| learning rate | 0.00005 |
| discount factor | 0.99 |
| clipping factor | 0.3 |
| entropy coefficient | 0.0 |
| batch size (Multiwalker) | 20000 |
| batch size (Pursuit) | 5000 |

(c) Ape-X

| Parameter | Value |
|---|---|
| Feedforward shape | $(64, 64)$ |
| activation function | tanh |
| replay buffer capacity | 2000000 |
| batch size | 200000 |

A.2   DETAILED RESULTS

A.2.1   SCALABILITY

Table 3: The detailed results of the scalability experiments in the Longroad environment. The mean sample and learn times are scaled to correspond to 1000 timesteps.

| 10 Agents: | GA | CMA-ES | OpenAI-ES | PPO | PPO-PS |
|---|---|---|---|---|---|
| mean sample time [s] | 0.4629 | 0.4133 | 0.4648 | 2.4477 | 1.3700 |
| mean learn time [s] | 0.0009 | 0.0305 | 0.0012 | 4.9640 | 15.7019 |
| **100 Agents:** | GA | CMA-ES | OpenAI-ES | PPO | PPO-PS |
| mean sample time [s] | 0.9560 | 0.8561 | 0.9607 | 15.1163 | 4.5312 |
| mean learn time [s] | 0.0056 | 0.3240 | 0.0091 | 33.3100 | 155.9862 |
| **1000 Agents:** | GA | CMA-ES | OpenAI-ES | PPO | PPO-PS |
| mean sample time [s] | 4.1388 | 3.3275 | 4.1291 | 147.7115 | |
| mean learn time [s] | 0.1135 | 4.6325 | 0.2196 | 323.6083 | |

Table 4: Detailed iteration timing results for OpenAI-ES

| agentsize | iteration time [s] |
|---|---|
| $10^1$ | 7.50 |
| $10^2$ | 1.05 |
| $10^3$ | 1.35 |
| $10^6$ | 573.71 |

### A.2.2 COMPERABILITY

Table 5: The detailed timing results of the comparability experiments.
The mean sample and learn times are for one iteration or scaled to correspond to 1000 timesteps.

| **Pursuit:** | | |
|---|---|---|
| | OpenAI-ES | PPO |
| mean sample time (iteration) [s] | 19.24971 | 17.90616 |
| mean learn time (iteration) [s] | 0.23939 | 62.51912 |
| mean sample time (1000 steps) [s] | 0.02135 | 0.31975 |
| mean learn time (1000 steps) [s] | 0.00027 | 1.11641 |
| **Multiwalker:** | | |
| | OpenAI-ES | PPO |
| mean sample time (iteration) [s] | 8.5248 | 5.75671 |
| mean learn time (iteration) [s] | 0.1296 | 28.18318 |
| mean sample time (1000 steps) [s] | 0.01213 | 0.22824 |
| mean learn time (1000 steps) [s] | 0.00018 | 1.11737 |

Table 6: The solution selection results for co-evolutionary OpenAI-ES for the best sampled fitness over all generations, the average fitness for the generation of the best sampled fitness and its evaluation over 32 episodes in parameter sharing.

| **Pursuit:** | | | | |
|---|---|---|---|---|
| | Trial 1 | Trial 2 | Trial 3 | Average |
| best fitness | 8.34898 | 4.09528 | 7.82374 | 6.75600 |
| population fitness of best fitness | 2.32692 | -8.23141 | -7.05302 | -4.31917 |
| evaluation of best fitness | -18.19020 | -9.51989 | -18.97909 | -15.56306 |
| **Multiwalker:** | | | | |
| | Trial 1 | Trial 2 | Trial 3 | Average |
| best fitness | 4.18716 | 4.94261 | 4.65052 | 4.59343 |
| population fitness of best fitness | -11.02063 | -0.38326 | -6.19596 | -5.86661 |
| evaluation of best fitness | 2.00721 | 3.70874 | -36.13141 | -10.13849 |

Table 7: Timings of parameter shared and co-evolutionary OpenAI-ES in the Multiwalker environment

| method | sampling time [s] |
|---|---|
| COMARL | 8.70872 |
| PS | 23.00535 |

