# OpenReview forum: "Co-Evolution As More Than a Scalable Alternative for Multi-Agent Reinforcement Learning"
_ICLR.cc/2023/Conference — Submitted to ICLR 2023_

### Official Review · Reviewer_d1BT · 2022-10-21

**Confidence:** 4
**Correctness:** 2
**Technical Novelty And Significance:** 2
**Empirical Novelty And Significance:** 2
**Recommendation:** 3

**Clarity, Quality, Novelty And Reproducibility:**

Please see above. The topic is quite interesting but the clarity of this paper should be considerably improved.
If I am correct, this paper is to integrate co-evolution with parameter-sharing. I think this would work and is novel. But current version of manuscript is too vague.
As it is difficult for me to understand the algorithm with no textual description, I am not sure whether it is reproducible.



**Strength And Weaknesses:**

Strength:
1. The topic of scalability of MARL is important and waiting to be addressed.
2. The co-evolution is intuitively suitable for this problem.
3. The empirical setup is clear, though it can be further improved.

Weaknesses:
The major drawback of this paper is the clarity of writing. There are several important points look very vague to me.
1. Though co-evolution is intuitively suitable, it is unclear what makes it really successful. Especially it is not the first MARL work based on co-evolution. Indeed, there have been lots of co-evolution based methods for MARL in the literature. Then why those methods fail to scale up MARL and why this proposed method can work successfully? This part has not been discussed clearly. And even many co-evolution based MARL methods have not been correctly referred in this work. To me, it looks like the success of COMARL is not from co-evolution, but from parameter sharing. Which one contributes more? This needs some empirical evidence.
2. The Algorithm 1 has not been clearly described. It is difficult to understand how exactly the COMARL works and thus it is impossible to judge its correctness and advantages.
3. Some empirical results have not been clearly discussed. For example, in Figure 1, the reason of not showing PPO-PS in 1000 agents case was given as "Because of the expected learning time consumption of PPO-PS for this setting". What does it mean? Too long running time? Then how long is it? To what extent of the running time will the algorithm be omitted from this comparison. As seen from Table 3, PPO-PS is even faster than PPO in the 10 agents and 100 agents cases. Also for Table 3, why the mean sample time of these algorithms grow much faster from 100agents to 1000agents than that from 10agents to 100agents? Last but not least, in Table 4, what the iteration time in 100agents and 1000agents is even much smaller than that in 10 agents, and the iteration time then quickly grow 570x in 1000000agents case? These points are rarely discussed or even mentioned in the paper, and they are important to judge the correctness and fairness of the empirical studies.

Another issue is about the longroad experiment where the COMARL is shown to scale up to 1000000 agents. Indeed, a scale of 1000000 agent is highlighting. But a comparison between COMARL and other SOTA methods on this scale will be more illustrative and convincing.

In the second paragraph of introduction, "a ensemble"->"an ensemble".


**Summary Of The Paper:**

This paper studies the central problem of the scalability of multi-agent reinforcement learning system (MARL). The proposed method is based on the co-evolution framework that is intuitively suitable for this problem. It is empirically verified with some benchmarks. The contribution is potentially a powerful co-evolution framework for MARL.

**Summary Of The Review:**

In general, this paper cannot be accepted at its current status. However, if it can be clarified significantly and correctly, I think this topic of using co-evolution for scaling up MARL  worth an ICLR paper.

---

### Official Review · Reviewer_da8i · 2022-10-23

**Confidence:** 3
**Correctness:** 2
**Technical Novelty And Significance:** 2
**Empirical Novelty And Significance:** 1
**Recommendation:** 3

**Clarity, Quality, Novelty And Reproducibility:**

- Clarity: Much of the text is very difficult to read and the overall contributions and methodology are hard to understand.
- Quality: Low. Experiments focus on a new toy domain and are hard to compare for previously used environments.
- Originality: Modest. Takes a reasonable approach to scaling coevolution of policies when populations exceed the number of agents in an environment at any one time.
- Reproducibility: Moderate. Code was provided in the supplement. Details are hard to follow from the paper, so reproduction would require running the code.

**Strength And Weaknesses:**

## Strengths

1. Novel MARL problem: supporting coevolution of agents while maintaining a large population.

## Weaknesses

1. Problem significance. It is not clear where a large population of potentially cooperating agents ties into MARL problems. Is the idea that agents must be able to cooperate with potentially diverse partners?
2. Early results: only tested scaling in the simplest environment (longroad).
3. Comparison issues in experiments. For example, the experiments should include PPO-PS in Figure 1c.
4. Text clarity. The overall narrative of the text is difficult to follow. The paper would benefit from first defining the problem being investigated and stating it's importance in the introduction. Then the related work can clarify alternative approaches. The experiments section would be clearer if the set of hypotheses to test were first defined, followed by the methodology for each experiment, followed by the results from the experiments.



## Feedback & Questions
- Figure 1: Would benefit from displaying variance of runs.
- Figure 4: The two subfigures should use the same axis scale to facilitate comparison.
- Results should align in terms of number of samples or wall clock time. Direct comparisons are hard to make as the data reported varies in both. What is being held constant?

**Summary Of The Paper:**

The paper studies the problem scaling coevolutionary algorithms to many actors in multi-agent reinforcement learning tasks. The paper introduces an algorithm that creates evaluation populations by randomly sampling members from a larger population to generate groups for each evaluation sampling. Sampling rates are balanced to ensure each member gets a minimum number of samples before being evolved. Experiments introduce a new environment that supports thousands of agents to test scaling and compare the algorithm to two baselines in another MARL domain.

**Summary Of The Review:**

I struggled to follow the overall claims of the paper, but my understanding is the paper is considering multi-agent reinforcement learning problems and proposing the use of evolutionary algorithms to optimize agents for these problems. From a technical perspective this is a modestly novel approach (evolutionary algorithms have been widely adopted in single agent RL problems and variants have been applied to competitive RL problems). Empirical results are relatively lacking: the longroad results discuss scaling but it was not clear what was being held constant across runs among the algorithms. The scaling outcomes (in wallclock time) are hard to interpret due to the large implementation differences in algorithms and parallelization. The other MARL domains show strong performance in one environment and reasonable performance in another, though only compared to a small set of alternatives. Taken together it is not clear that the paper has backed it's core claims. The work will be stronger if the results could show directly comparable implementations using a broader set of MARL environments and algorithms.

---

### Official Review · Reviewer_3o7H · 2022-10-24

**Confidence:** 3
**Correctness:** 1
**Technical Novelty And Significance:** 1
**Empirical Novelty And Significance:** 1
**Recommendation:** 1

**Clarity, Quality, Novelty And Reproducibility:**

*Clarity*: Poor. The paper lacks simple clarity and is difficult to follow and make sense of.

*Quality*: Poor, considering many weaknesses, it has low quality, way beyond the expected level of ICLR.

*Novelty*: Modest, integrating co-evolution with parameter-sharing is novel, as far as I'm concerned.

*Reproducibility*: Modest. Although the authors attached their code, algorithm's details are difficult to understand from the manuscript.

**Strength And Weaknesses:**

Unfortunately, the paper does not have any strengths.

Overall, it is very unclear and incoherent.

- Most of the text is very difficult to read and understand. Hence, the overall contributions and methodology are difficult to grasp.
- The paper doesn’t have Background and Related Work sections, which is really surprising considering it is only 8 pages long (two of which are just graphs and mostly empty).
- The description of the method in Algorithm 1 is very unclear. The authors did not introduce any of the notations used in the description. It is unclear what are $F$, $\theta$. $\mathcal{F}$, etc.
- The experimental results also do not make much sense. According to the authors, some environments are not there to “showcase any sophisticated learning behaviour but rather to be a primitive test bed for testing the large scale MARL concepts.” Furthermore, this environment (called Longroad) “is not a good benchmark in the general setting”. Rather than focusing on new toy domains, the authors should have compared with previously used environments.
- Plots in Figure 1 only seem to have a single run per method without any confidence intervals. Some of the runs in Figures 1-3 abruptly end half-way-through the training with an unclear final performance.

**Summary Of The Paper:**

The main idea behind the paper seems to be handling a large number of agents in multi-agent reinforcement learning (MARL). The authors provide a description of their method based on co-evolution framework. They provide some experimental results for showing the scaling in the number of agents.


**Summary Of The Review:**

This paper lacks basic clarity and coherence. The motivation of the work is unclear and the method is not properly described. Experimental results also do not make much sense and have many flaws.

---

### Official Review · Reviewer_9b7B · 2022-10-26

**Confidence:** 5
**Clarity, Quality, Novelty And Reproducibility:** This paper is poorly written and not …
**Correctness:** 1
**Technical Novelty And Significance:** 1
**Empirical Novelty And Significance:** 1
**Recommendation:** 1

**Strength And Weaknesses:**

This paper is poorly written, not well-motivated, has insufficient related work, a confusing algorithm, and weak experimental results.

**Summary Of The Paper:**

This paper proposes a co-evolutionary algorithm to handle some issues from large-scale multi-agent RL. The proposed method does not only consider sampling in concurrent environments but further investigates sampling diverse parameters from the population in co-evolution in joint environments during training.

The experimental results show that it is capable of training a large number of agents.

**Summary Of The Review:**

1. The structure of the paper is badly organized.

2. The motivation is very unclear.

3. The related work of this paper is very limited. As a manuscript submitted to a top-venue conference in 2022, there are too few recent papers cited in this paper.

4. As a paper to study multi-agent RL, none of the multi-agent RL algorithms have been discussed and compared in the experiments, e.g., VDN and QMIX.

---

### Decision · Program_Chairs · 2023-01-20

**Decision:**

Reject

**Justification For Why Not Higher Score:**

Not a good paper.

**Justification For Why Not Lower Score:**

N/A

**Metareview: Summary, Strengths And Weaknesses:**

This paper discusses co-evolution as an alternative to MARL. Unfortunately, it has multiple flaws, including lack of novelty, unconvincing results, insufficient experimentation, lack of recent related work, and unclear writing.